# Improving Access to Cancer Treatment Services in Australia’s Northern Territory—History and Progress

**DOI:** 10.3390/ijerph19137705

**Published:** 2022-06-23

**Authors:** Emma V. Taylor, Rosalie D. Thackrah, Sandra C. Thompson

**Affiliations:** Western Australian Centre for Rural Health, The University of Western Australia, Geraldton, WA 6530, Australia; rosalie.thackrah@uwa.edu.au (R.D.T.); sandra.thompson@uwa.edu.au (S.C.T.)

**Keywords:** cancer, indigenous, remote, Northern Territory, Australia, palliative care, health services, treatment access

## Abstract

Cancer is the leading cause of death in the Northern Territory (NT), Australia’s most sparsely populated jurisdiction with the highest proportion of Aboriginal people. Providing cancer care to the NT’s diverse population has significant challenges, particularly related to large distances, limited resources and cultural differences. This paper describes the developments to improve cancer treatment services, screening and end-of-life care in the NT over the past two decades, with a particular focus on what this means for the NT’s Indigenous peoples. This overview of NT cancer services was collated from peer-reviewed literature, government reports, cabinet papers and personal communication with health service providers. The establishment of the Alan Walker Cancer Care Centre (AWCCC), which provides radiotherapy, chemotherapy and other specialist cancer services at Royal Darwin Hospital, and recent investment in a PET Scanner have reduced patients’ need to travel interstate for cancer diagnosis and treatment. The new chemotherapy day units at Alice Springs Hospital and Katherine Hospital and the rapid expansion of tele-oncology have also reduced patient travel within the NT. Access to palliative care facilities has also improved, with end-of-life care now available in Darwin, Alice Springs and Katherine. However, future efforts in the NT should focus on increasing and improving travel assistance and support and increasing the availability of appropriate accommodation; ongoing implementation of strategies to improve recruitment and retention of health professionals working in cancer care, particularly Indigenous health professionals; and expanding the use of telehealth as a means of delivering cancer care and treatment.

## 1. Introduction

Over the last two decades, advancements in cancer treatments have resulted in increased cancer survival rates for most developed nations [1,2,3]. However, these improvements are not shared equally within populations, with significantly poorer cancer outcomes reported for people living in rural and remote areas, Indigenous peoples and people from low socio-economic backgrounds [4,5,6,7,8]. These disparities persist despite increased attention and collaborative international efforts to understand and reduce these variations [9].

Australia leads the world in cancer survival rates [2]; however, Australians living in rural and remote areas are known to experience poorer survival outcomes from cancer, with cancer survival rates generally decreasing as remoteness increases [10,11]. Australia’s Aboriginal and Torres Strait Islander peoples represent 3.3% of the total Australian population and have richly diverse and complex cultures [12] (hereafter the term “Indigenous Australians” is respectfully used); however, systemic discrimination and disadvantage dating back to colonisation have resulted in Indigenous Australians experiencing lower levels of education and employment, shorter life expectancy and worse outcomes across a wide range of diseases including cancer [13,14]. While 61% of Indigenous Australians reside in major cities or inner regional areas, they comprise a greater proportion of the total population as remoteness increases, with Indigenous people comprising 1.7% of the population in major cities, 18% of the population in remote areas and 47% in very remote areas [12,15]. Indigenous patients from rural and remote locations face additional challenges of distance and cultural barriers when accessing specialist care [16,17,18,19]. Therefore, Indigenous Australians living in rural and remote areas experience a “double dose of disadvantage” when it comes to disparities in cancer outcomes and have higher rates of hospitalisation and cancer mortality than Indigenous people living in urban areas [20,21].

The Northern Territory (NT) is the third largest of Australia’s states and territories and the 11th largest country subdivision in the world [22]. With a population of 246,700, the NT has only 1% of the national population [23], giving it the lowest population density of any Australian jurisdiction (0.2 people per km^2^) [24]. The NT population has a number of unique characteristics, including a relatively young population (median age 33 years compared to the national median age of 37 years); the highest ratio of males to females (107:100); and the highest proportion of Indigenous people (30% of the total NT population and 9% of Australia’s total Indigenous population) [25]. The geographic distribution of the population is distinctive: the majority of Indigenous residents (78%) live in remote areas, while the majority of the non-Indigenous population (75%) live in the capital city of Darwin [26]. The NT population is culturally and linguistically diverse, with almost 30% of the population speaking a language other than English at home [27]. In addition to migrants and workers from many other countries, the NT has one of the most diverse Indigenous populations in Australia, with over 100 different Indigenous languages spoken throughout the Territory and 15% of the population speaking an Indigenous language at home [28]. In addition, 20% of the population were born outside Australia, and around 160 foreign languages are also spoken in the home [28,29]. In the most recent census, educational attainment was below the national average, with 53% of the population (aged over 15) completing school (year 12 or equivalent) compared to a national average of 62% [27]. However, the unemployment rate is lower than the national average (4% compared to 5.4%), and a much higher proportion of the population is engaged in full-time work (82% compared to 68%) [23].

Cancer is the leading cause of death in the NT, accounting for 33% of all non-Indigenous deaths and 17% of all Indigenous deaths between 2006 and 2014 [30]. Although the NT has a lower incidence rate of cancer compared to Australia as a whole, the age-standardised rate of cancer mortality is higher, particularly for the NT Indigenous population [30]. One probable contributing factor is access to services, with the NT having just 1% or three of Australia’s cancer services [31], with remoteness and low population density presenting major challenges for health service and particularly cancer treatment delivery. A high prevalence of smoking is another contributing factor, with the NT population historically having the highest smoking prevalence in Australia (in 2019, 14.7% compared to the Australian prevalence of 11%), and an Indigenous smoking prevalence of 45%, the highest of all states and territories [32,33]. As with Australia generally, the gap between Indigenous and non-Indigenous cancer mortality is widening, with the cancer mortality rate among the non-Indigenous population steadily declining, while for Indigenous peoples, cancer mortality has increased by 1% per year between 1991 and 2015 [30]. Indigenous people diagnosed with cancer in the NT are more likely to be diagnosed at a more advanced stage, and less likely to attend medical appointments prior to radiation therapy sessions and to decline the recommended radiotherapy treatment [34]. Furthermore, the incidence of several high fatality cancers, including liver, lung and other smoking-related cancers, is much higher in Indigenous than in non-Indigenous people, and the incidence of breast, bowel and prostate cancers is increasing (historically, incidence was lower in the NT Indigenous population), necessitating attention on improved services for screening, diagnosis and treatment for this group [30,35].

Providing cancer care to the Northern Territory’s diverse population has significant challenges, especially related to large distances, limited resources and cultural differences. However, there have been major developments in the delivery of cancer services in the NT over the last two decades, with more local resources available for early detection and treatment. This paper describes these historical developments and outlines the demographic and geographic considerations of care provision in the NT, with a particular focus on the NT’s Indigenous peoples. It describes the resources currently available for the care of NT cancer patients, and treatments with radiotherapy, chemotherapy and surgery as well as screening and end-of-life care. These developments in the management of patients, from screening to cancer treatment through to support for cure or palliation, contribute to an understanding of the history of health service funding and delivery in Australia and also have relevance for countries with substantial rural or remote populations or with similar histories of colonisation and marginalisation of the Indigenous population, including Canada, New Zealand and the United States of America (USA).

## 2. Methods

This study is a component of research that was undertaken as part of a Centre for Research Excellence (CRE), Discovering Indigenous Strategies to improve Cancer Outcomes Via Engagement, Research Translation and Training Centre of Research Excellence (DISCOVER-TT). The CRE was led by an Indigenous researcher and involved Indigenous and non-Indigenous people working together to improve services and outcomes for Indigenous people with cancer. Ethics approvals for the national study were obtained from the Western Australian Aboriginal Health Ethics Committee (WAAHEC) (approval number 483) and the Human Research Ethics Committees of University of Western Australia (RA/4/1/6286).

This overview of the development of cancer services in the Northern Territory forms part of a broader investigation to identify and describe cancer services providing treatment to Indigenous cancer patients in Australia. During the national study, a cancer service in the NT was identified as providing innovative services to support their Indigenous patients and agreed to participate in a case study. However, while undertaking the case study it was apparent that there was little information around cancer service development in the NT, or at least it was difficult to find and scattered, so this paper was developed to address that gap. In addition, the NT was considered to be of particular interest to other rural and remote areas struggling with cancer service provision given the recency of its development of services, especially in rural and remote locations.

This overview of the development of cancer services was collated from peer-reviewed literature, government reports, cabinet papers and personal communication with health service managers and providers. Peer-reviewed literature was identified through searches conducted in January 2020 and January 2022 of electronic databases: PubMed, Informit and Google Scholar. Key search terms “cancer” and “Northern Territory” were searched using a combination of prescribed subject headings and free-text keywords. Additional searches were run with the search terms “Indigenous” and “Aboriginal”, as well as prescribed subject headings, to identify Indigenous-specific articles. To identify relevant grey literature, the websites of the following organisations were searched: Northern Territory Department of Health, Northern Territory Primary Health Network (NT PHN), Australian Institute of Health and Welfare (AIHW) and NT hospital websites. Citation snowballing was also utilised.

## 3. Results

### 3.1. Description of Overall Service Delivery

Within Australia, state and territory governments have responsibility for providing public hospital services. The NT Department of Health has divided the Territory into two regional government health services—the Top End Health Service and the Central Australia Health Service. These regions are further divided into sub-regions (Figure 1). There are major public teaching hospitals in Darwin and Alice Springs, as well as smaller public hospitals in Palmerston, Gove, Katherine and Tennant Creek, and one private hospital (Darwin Private Hospital).

Until the early 1980s, specialist services in the Northern Territory were limited, and services were either provided by visiting medical specialists, usually from South Australia, or by transferring patients interstate [36,37]. In Central Australia, interstate patient transfer is still a frequent occurrence, with patients transferred “when highly specialized care is required … to major hospitals in Adelaide and beyond” [38].

The NT population has grown from 192,700 in 1999 [39] to 244,800 in 2019 [40], and in response new health services have been established. Most cancer services are now available on the Royal Darwin Hospital (RDH) campus, with more limited services available in Alice Springs and Katherine. The Alan Walker Cancer Care Centre (AWCCC), located on the grounds of RDH and opened in January 2010, is a purpose-built radiation oncology and outpatient chemotherapy facility that services the whole of the NT, providing outpatient chemotherapy and radiotherapy services within a public–private business model between Northern Territory Radiation Oncology and NT Department of Health [34,41]. Patients who reside outside the Darwin region are required to relocate to Darwin for the duration of their radiotherapy treatment. Courses of curative radiotherapy treatment are usually provided five days a week over several weeks (or months). Treatments for paediatric patients and some complicated cancers are not available in the NT, requiring these patients to relocate interstate, with their escort, for extended periods of time. An outline of current availability of services in various locations across the NT in 2022 is shown in Table 1.

### 3.2. Top End Cancer Services

The Top End Health Service (TEHS) region covers 475,338 km^2^, approximately 35% of the total area of the NT, and includes the Darwin, East Arnhem and Katherine health districts. In June 2020, the TEHS region had an estimated resident population of 200,450 people, representing 82% of the total NT population [25].

The RDH campus is a multiuser site, and co-locates the facilities and services of RDH, AWCCC, Territory Palliative Care (TPC) and Darwin Private Hospital. RDH is a university teaching hospital, incorporating the Menzies School of Health Research and Flinders Medical School, providing a range of services in all specialty areas to the Darwin urban population as well as serving as the referral centre for the whole Top End of the NT. Services include a 24-h emergency department, critical care services and medical imaging including nuclear medicine. The AWCCC provides outpatient chemotherapy and radiotherapy services to NT patients. The chemotherapy suite has 16 chemotherapy chairs, increasing to 20 by 2024 [41]. Additional services provided by the AWCCC include haematology, palliative care, pathology, clinical trials and telehealth. A range of support services and allied health services are also available, including cancer care coordination, Indigenous liaison support, social work, physiotherapy, occupational therapy and dietetics.

The RDH and Siemens Healthineers partnership expanded the NT’s oncology services in 2019 with the addition of the Positron Emission Tomography (PET) scanner and the associated cyclotron facility [43] to significantly reduce the need for NT patients to travel interstate and enable improvements in patient outcomes through earlier diagnosis and more detailed treatment plans. The PET scanner provides the capability to scan faster with lower doses of radiation and with scans individualised to the patients’ medical condition [43]. Previously, patients travelled interstate to access PET scanning equipment for diagnosis and treatment monitoring [43].

Katherine is a town 320 km southeast of Darwin, with a population of 6300 [44]. Katherine Hospital serves as a referral centre for the surrounding remote communities. Although Katherine Hospital has a limited cancer diagnostic capacity, with patients travelling to Darwin for investigations and diagnosis, in 2015 it opened a chemotherapy unit with four chemotherapy chairs, allowing eligible patients to receive chemotherapy treatment closer to home [41,45].

### 3.3. Central Australian Health Service

The Central Australia Health Service (CAHS) region covers 872,861 km^2^, almost two-thirds (64.7%) of the total area of the NT and servicing a population of 45,000. The CAHS includes the Alice Springs and Barkly Health Districts, and a number of Indigenous communities [25].

Alice Springs Hospital (ASH) is the referral hospital for Central Australia with a catchment area extending into the tri-state border areas of South Australia (SA) and Western Australia (WA). ASH provides a range of general secondary and some tertiary inpatient and outpatient services. Medical, surgical and radiation oncology specialists provide cancer management by visiting ASH from Darwin and Adelaide on a regular basis, supplemented with telehealth consultations and multidisciplinary team (MDT) meetings. Between 2013 and 2016, ASH experienced a 50% increase in chemotherapy activity, indicating demand for additional services [41]. In 2018, ASH opened a new six-chair chemotherapy day unit with access to a range of allied health and support services [41].

### 3.4. Workforce

Health professionals are the human faces of any health system [46]. In 2018, the NT had the highest number of registered health professionals relative to its population (2790 FTE per 100,000 people) of all Australian jurisdictions [47]; however, most of the health workforce is based in the capital city of Darwin, and access to health professionals outside the urban centres is more limited. The NT has the lowest per capita rate of specialists and dental practitioners and the second lowest rate of allied health practitioners [47,48]. See Table 2 for a breakdown of the NT health workforce by profession and region. Furthermore, health professionals were predominantly younger, less experienced, overseas-trained and employed on short-term, agency or locum contracts [49,50,51]. The NT Department of Health and the NT Primary Health Network (PHN) report challenges in recruiting and retaining health professionals, particularly medical generalists, specialists and allied health professionals, especially in rural and remote areas [25,49,50,52]. A study of workforce turnover and retention in remote communities found very high annual turnover rates of Remote Area Nurses (148%), with only 20% of nurses and allied health professionals remaining at the same remote clinic 12 months after commencing, and half leaving within 4 months [51]. Workforce recruitment and retention were also identified as issues in the NT Cancer Care Strategy 2018–2022 [41].

Indigenous people are underrepresented in the NT health workforce, with only 8.1% of the Top End Health Service (TEHS) and 7.1% of the FTE workforce employed by the Central Australia Health Service (CAHS) identifying as Indigenous [25]. Only 21 allied health professionals identified as Indigenous in 2019 [49,55], and only 2.2% of nurses and 2.3% of General Practitioners identified as Indigenous in the NT [50]. Aboriginal Health Practitioners (AHPs) and Aboriginal Health Workers (AHWs) play a critical role in the delivery of health services to Indigenous people, particularly in remote areas of the NT [50]. Although data suggest this workforce is local and experienced, with lower turnover rates than nurses, there were only 226 AHPs in the NT in 2015, and numbers were declining [50,52]. Increasing the size of the Indigenous workforce working in cancer care is one of the nine evaluation indicators in the NT Cancer Care Strategy 2018–2022 [41].

### 3.5. Screening

There are federal government-funded population-based screening programs operating in the NT for the prevention or early detection of three types of cancer: breast and cervical cancer in women, and bowel cancer for men and women [56].

The National Bowel Cancer Screening Program (NBCSP) invites eligible people without symptoms to screen for bowel cancer using a simple test kit at home. Regular screening by a faecal occult blood test (with follow-up diagnostic colonoscopy after a positive screening test), can detect bowel cancers early so they can be removed while still small and localised. The NBCSP operates by sending screening kits to people every second year between the ages of 50 and 74 years. Although bowel cancer is among the most common cancers for the NT non-Aboriginal population and has increased considerably over the past three decades for the NT Aboriginal population, bowel cancer screening participation is lower in the NT than elsewhere in Australia, and lower for Aboriginal than non-Aboriginal people within the NT [56]. Among those who participated in screening, Aboriginal people had a higher prevalence of a positive screening result but were less likely to be followed up by diagnostic assessment after a positive result and less likely to receive treatment after cancer was diagnosed [56].

Australia has had a national coordinated breast cancer screening program (BreastScreen) since 1991 offering free mammograms every two years to women aged 50–74 years. Although breast cancer is the most common cancer for females in the NT, breast screening participation is lower for NT women than for Australian women generally and lower for NT Aboriginal women than non-Aboriginal women within the NT (39.8% for non-Aboriginal and 26.7% for Aboriginal women compared to 52.3% nationally) [56]. However, participation has increased for Aboriginal women over recent years, with the provision of a mobile breast screening bus service contributing to this increase [56]. For women needing assessment after a positive screening test or treatment after a breast cancer diagnosis, time to assessment and treatment was longer for Aboriginal than non-Aboriginal women [56].

In 1991, Australia introduced a national coordinated cervical screening program. Until November 2017, screening was carried out by the Pap test, an indirect test for changes in cervical cells caused by the HPV virus that was recommended every two years for women aged 20–69 years. Cervical screening participation in the NT is the lowest in the country (51.8% compared to 56%); however, participation for Aboriginal women has increased over recent years and is now almost the same as for non-Aboriginal women [56]. Although cervical cancer incidence has decreased by more than 50% between 1991 and 1995 and 2011 and 2015, indicating the effectiveness of the cervical screening program in the NT, the incidence of cervical cancer remains 60% higher for Aboriginal than non-Aboriginal NT women [56].

### 3.6. Telehealth

The NT has seen significant uptake of telehealth since the launch of the Health eTowns Program in 2010, developing facilities, installing a high-speed communications network and providing online telehealth training to health staff [57,58]. During COVID-19, the use of telehealth has accelerated [57]. One influencing factor is the NT’s significant challenges to accessibility and delivery of health care due to its vast size, small population, seasonal weather restrictions and limited number of urban centres. Telehealth services can assist in the delivery of health services to remote communities by reducing the need for travel, providing timely access to services and specialist doctors, improving the ability to identify developing conditions and providing a means to educate, train and support remote health care providers [59]. An evaluation of telehealth services in the NT found that it significantly improved clinic attendance rates, reduced Did Not Attend (DNA) rates and reduced travel costs [60].

As early as 2000, a videoconference link between the Royal Adelaide Hospital in South Australia and RDH was being used to discuss cases in multidisciplinary oncology meetings [61]. Since its official initial launch in the NT, telehealth has been used to deliver tele-oncology patient reviews [58]. Telehealth consulting supports the delivery of specialist oncology services to regional health services such as Alice Springs Hospital and Katherine Hospital [41]. Telehealth appointments enable specialist clinicians to provide consultations and follow-up reviews to cancer patients in their community where they can be accompanied by their traditional supports; telehealth is also used for family meetings to keep remote family members informed and involved. Using telehealth prior to patient travel and treatment aims to ensure that patients and their family understand the requirements of their cancer treatment, as the treatment can often require patients to be away from home for many weeks or even months. Telehealth is used for MDT meetings to ensure best-practice cancer treatment planning and coordination for NT cancer patients by engaging with Territory-wide and interstate specialist cancer care and primary health professionals. Palliative care services are now also being supported by telehealth [62].

Access to high-speed broadband internet is imperative for effective telehealth consultations; however, internet access and speeds in rural and remote Australia, including the NT, continue to be problematic [59]. Poor internet connection speeds and limited equipment, especially in remote areas, are some of the main hinderances to the growth of telehealth in the NT [60,62]. Training for health service providers on how to effectively use telehealth linking remote and regional health services with the specialist services is available; however, with the high turnover of NT health staff in rural and remote locations, expertise in delivery of cancer care by telehealth may be limited in practice [62,63].

### 3.7. Assisting Patients with Travel and Accommodation

Large distances and a lack of transport contribute to poorer access to appropriate primary, specialist and follow-up health care for rural and remote Australians [15]. Logistical and financial challenges for patients and providers may delay cancer diagnosis and management [64]. The Patient Assistance Travel Scheme (PATS) provides a financial subsidy that covers a portion of the “out of pocket” expenses incurred when eligible NT residents travel to approved specialist medical services. In the NT, the PATS provides assistance with transport and accommodation costs for rural and remote patients traveling more than 200 km from home to the hospital for treatment [65]. NT has many communities within 200 km of regional towns where people requiring cancer treatments must find and finance their own transport to attend their appointments. Telehealth can and has been used to reduce the necessity to travel for specialist medical appointments, but it is not appropriate for all consultations.

The distances to travel for specialist care and cancer treatment from some communities in the NT are vast, and very tiring for people undertaking investigations for diagnosis of cancer or undergoing cancer treatment. For example, the distance from Borroloola (a remote fishing town near the Gulf of Carpentaria) to Darwin is 974 km; it requires an 8½ hour bus journey from Borroloola to Katherine, an overnight stay and a further 4-h bus journey to Darwin. The weather conditions in the NT vary between the Top End and Central Australia but include extremes, such as the freezing cold of winter in Central Australia and searing desert heat in summer; Top End weather includes tropical heat, monsoonal rain and occasional cyclones in the wet season (October to April). These conditions mean journeys can be challenging, especially for people unwell with cancer and undergoing treatment. If health service providers believe that the patient and their escort require air transport, then they can negotiate with the PATS personnel who are responsible for determining eligibility.

The PATS provides an accommodation subsidy for eligible patients, including those with cancer [65]. Depending on the nature of the treatment plan, cancer patients and their carers may need to remain away from home for lengthy periods. In Darwin, the Barbara James House provides low-cost, short-term, catered accommodation for all rural and remote NT cancer patients accessing treatment [66]. It is the only accommodation exclusively available for cancer patients in the NT and is available for both Indigenous and non-Indigenous patients. In addition, a free shuttle service is provided from the Barbara James House to the AWCCC. Outside Darwin, there is a shortage of low-cost accommodation to support patients travelling for cancer treatment [67]. Furthermore, patients staying in commercial accommodation are responsible for any gap in costs. Accommodation costs can be problematic for patients who are not eligible for the PATS subsidy.

### 3.8. Palliative Care

Territory Palliative Care (TPC) provides palliative care services across the NT, with teams based in Darwin and Alice Springs. TPC aims to provide “high quality care that responds to the physical, psychological, social and spiritual needs of patients with an underpinning of cultural sensitivity” [41]. TPC provides services for inpatients (hospital and hospice) and an outpatient service for patients in the community. Top End TPC is the sole provider of palliative care services to NT’s Top End [68]. The 12-bed stand-alone palliative care unit/hospice on the RDH campus opened in 2004 [69], and is co-located with the rural and remote palliative care team. A patient transport vehicle is available if patients wish to return home “even if they live 500 km or more away” [70]. The Central Australian TPC provides inpatient support to patients in ASH and Tennant Creek hospital, as well as outpatient and community support to patients at home (urban or remote) or in aged care facilities [41]. The 10-bed Apmere Amantye-Akeme Palliative Care Facility (Comfort House) opened in 2018 on the ASH campus, and is the first dedicated inpatient end-of-life and respite care facility in Central Australia [41].

A key consideration in the NT when providing palliative care to Indigenous people from rural or remote areas is place of death, with many wishing to “finish up” (a culturally appropriate reference to death and dying) on Country (ancestral community and lands) [71,72]. Some Indigenous patients choose to forego life-prolonging treatments in tertiary health centres in order to go “home to Country” [71]. Reasons for this include the importance of their connection to land and community, spending time with and being cared for by family and passing on sacred knowledge to family [71,72]. Finishing up on Country usually takes place in the patient’s own home or the home of a relative or close community member who has assumed responsibility for their care [68].

## 4. Discussion

This paper provides an overview of the challenges of delivering cancer care in the NT as well as developments to improve cancer treatment services over the past two decades, with a particular focus on what this means for the NT’s Indigenous peoples. The NT covers a large geographical area; its small, diverse population distributed across many small settlements requires efforts to reconceptualise traditional services and deliver more culturally appropriate care closer to home. Recent years have seen the NT make considerable progress in the development of cancer care services to help address some of these issues. Of particular note is the establishment of the AWCCC, which has contributed to a significant increase in Indigenous patient attendance for radiation therapy [34], the recent investment in a PET Scanner, the new six-chair chemotherapy day unit at ASH, the new four-chair chemotherapy day unit at Katherine Hospital and the rapid expansion of tele-oncology. There has also been a marked improvement in access to specialist palliative care facilities, with end-of-life care now available in Darwin, Alice Springs and Katherine.

Distance has a detrimental impact upon patients’ willingness to take up cancer treatment and impacts overall survival [73,74], so these additional facilities and services whereby NT residents can remain within the Territory for most cancer diagnoses and treatments are likely to have positively impacted treatment options and survival for NT patients. Improvements in cancer mortality are occurring; however, these may be offset by increases in incidence in Indigenous Territorians, with a concomitant effort to mitigate lifestyle risks required [35]. Furthermore, just having the “bricks and mortar” services is not enough to guarantee engagement by Indigenous people, as cancer treatment services are largely designed to suit the comfort and cultural norms of dominant groups who are more committed to Western medicine and more comfortable with prevailing hospital systems of care [75]. Lack of connection to Country, absence of local family support, mistrust of mainstream health services, lack of respect or cultural understanding shown by staff, and communication difficulties have all been described by Indigenous patients with cancer [76,77], including in the NT [78]. This can result in a poorer understanding of the treatment options available, difficulty obtaining fully informed consent to the cancer treatment process and lower uptake of cancer treatment [79,80,81,82]. While much can be done to build and create safe physical environments for the care of Indigenous people undergoing cancer treatments [83], previous work has highlighted that it is the cultural and social support provided by Indigenous staff, as well as culturally appropriate, patient-centred care from non-Indigenous staff, that are of overwhelming importance [34,75,84].

Increasing the number of Indigenous health professionals working in cancer care in the NT and improving the understanding of cultural beliefs and practices of non-Indigenous staff are critical to improving the cultural safety of cancer care in the NT [78]. Furthermore, while the NT had a higher number of registered health professionals relative to its population, this is offset by much higher rates of turnover as well as greater numbers of staff on short-term contracts, and higher use of agency-employed staff and locums than other Australian jurisdictions [85,86]. An NT study found that high turnover impacts cultural safety and continuity of care and leads to poorer health outcomes for Aboriginal peoples and higher average health costs [85]. In addition, the NT workforce overall is younger, less experienced and unevenly distributed, with much of the workforce located in the urban centres. While increased funding has been shown to help with workforce issues in the short term, supply that was related to financial boosts tends to dissipate over time, suggesting that additional strategies are required [52]. For non-Indigenous staff, strategies might include training to work in a remote setting, additional management and clinical support, training on communicating with people from culturally and linguistically diverse backgrounds, professional development opportunities and community programs to help health professionals build social connections [52,87]. For Indigenous workers, research suggests that opportunities for career progression and professional development are needed, particularly for AHPs and AHWs. Strategies recommended aim to address heavy workloads and burnout, support balancing work and community responsibilities, provide additional clinical and cultural mentoring and build supportive and culturally safe workplaces [52,88].

Darwin’s purpose-designed Barbara James House provides safe accommodation for all NT cancer patients and their escorts; however, in other areas of the NT, including Alice Springs and Katherine, accommodation remains limited, and not all patients are eligible for the PATS. Travel burden is a significant concern for many rural and remote cancer patients; it is associated with greater financial burden and may affect decisions about whether to undergo or continue with cancer treatment [89,90,91]. Travel assistance schemes, such as the PATS, are essential for rural Australians to be able to access cancer treatment [90]; however, this is problematic for patients who do not fit the PATS funding guidelines, such as people in the NT living within 200 km of a regional town. The NT’s distance-to-treatment criterion is greater than for other Australian jurisdictions, with Queensland patients eligible if travelling more than 50 km [92] and WA patients eligible if travelling more than 70 km for cancer or renal treatment [93]. An Australian government review into the PATS concluded that the 200 km eligibility limit imposed on NT residents makes “little sense in a jurisdiction where a relatively high proportion of the population live in communities with unsealed road access, no public transport, and limited flight services” and has been described as “a product of history rather than responsive policy” [94]. Even if a patient is eligible for the PATS, a recent NT study found many Indigenous patients experienced difficulties using the scheme, finding it confusing and inflexible, as well as experiencing substantial challenges with transport and unsuitability of accommodation [78].

The increasing utilisation of telehealth in the provision of cancer care has helped reduce the travel and financial burden on patients, while improving service provision and access, particularly for remote communities, and achieving cost savings for the health care system [60,95,96]. Studies have found telehealth to be acceptable to rural and remote patients, including Indigenous patients and cancer patients, with reports of improved social and emotional wellbeing, access to health services and clinical outcomes [84,97,98,99]. A regional cancer service in Queensland has successfully used telehealth-supported chemotherapy services at remote satellite sites with patient benefits including reductions in travel costs, travel stress and waiting time, plus family being included in consultations and treatment plan discussions [100].

Improvements in cancer outcomes for Indigenous Australians are related to the efforts to improve cancer diagnosis and treatment services, as well as improvements in cancer screening. The Australian government responded to criticism that the original national bowel cancer screening program was not appropriately designed to encourage uptake by Indigenous Australians [101,102,103,104], and trials of alternative approaches involving primary care services in the distribution of faecal occult blood screening test kits are underway [105]. Following the successful implementation of the HPV vaccine in 2007, Australia’s cervical screening program changed to a five-yearly human papillomavirus (HPV) test in December 2017, with the introduction of self-collection for eligible participants [106]. It is hoped that the increased testing interval will reduce the effort involved to participate in screening for rural and remote women and that self-collection will make screening more culturally acceptable for some Indigenous women [106]. Despite these changes to cancer screening, participation in screening programs in the NT remains significantly lower than in other areas of Australia [56]. Low uptake may be due to several reasons, including the large distances required to travel to screening sites, high turnover of health professionals, staff workload in remote areas, lack of knowledge of screening requirements (cervical and bowel screening) [106,107,108] and competing health priorities of the patients, including high rates of chronic conditions [109], caring responsibilities [110] and mental health problems [111,112].

There are ongoing efforts to improve the reporting of Aboriginal and Torres Strait Islander cancer data as part of the implementation of the *National Aboriginal and Torres Strait Islander Cancer Framework* [113]. The Aboriginal and Torres Strait Islander Cancer Control Indicators website is under development [114]. It is hoped that this will enable, over time, better mapping of the extent to which efforts to improve cancer services in the NT are successfully reducing the gap in cancer outcomes for Indigenous people living the NT compared to other Territorians and Indigenous Australians living elsewhere in Australia.

### 4.1. Limitations

While we have made every effort to gather an accurate and complete picture of the historical development and current overview of cancer services in the Northern Territory, describing these historical developments is challenging, as public announcements of initiatives do not instantly translate into services and there is no authoritative repository of policy and service delivery changes. In fact, half of the references in this article are in the grey literature. Furthermore, services may have initiatives that were not identified or reported. Service delivery is dynamic, while data collection for this article occurred over a discrete period. COVID-19, for example, has had a massive impact on cancer service delivery world-wide, including in the NT; however, until November 2021, the Northern Territory was successful in preventing community transmission, and up until January 2022, the Territory had one of the lowest rates of COVID-19 in Australia. It is therefore too early to describe the impact this has had in the NT.

### 4.2. Implications

It is hoped that findings from this article will highlight opportunities for the NT Government and NT Health Service to:Improve travel assistance and support, including review of the 200 km eligibility limit for the PATS.Increase the availability of appropriate accommodation for patients travelling for cancer treatment, including in Alice Springs and Katherine.Develop additional strategies to recruit and retain the cancer workforce and remote workforce:
Non-Indigenous health workforce: training to work in a remote setting, additional management and clinical support, professional development opportunities and community programs to help health professionals build social connections.Indigenous health workforce: providing additional opportunities for career progression and professional development, strategies to address heavy workloads and burnout, support for balancing work and community responsibilities, provide additional clinical and cultural mentoring and build supportive and culturally safe workplaces.Develop further localised strategies to increase participation in screening programs—including increased education for remote health professionals. Localised screening programs should be collaboratively designed with the community to make them acceptable to Indigenous people and people in remote areas.

## 5. Conclusions

The developments described show that the NT has succeeded in advocating for cancer patients and their families by increasing the cancer care infrastructure and services, allowing patients to stay closer to home and Country, improving access to specialist expertise and improving cancer outcomes. These important resources include cancer treatment facilities, provision of cancer care via telehealth and improved access to accommodation. Concurrent enhancement of the palliative care facilities in the NT’s major urban centres reflect considerations of end-of-life care, a particularly important issue for Indigenous Australians. However, future efforts in the NT should focus on increasing and improving travel assistance and support, including reviewing the 200 km eligibility limit for the PATS, and increasing the availability of appropriate accommodation. Research on effective strategies to improve the recruitment and retention of health professionals working in cancer care, particularly Indigenous health professionals and health professionals working in remote settings, is needed. Additional research into the effectiveness and appropriateness of telehealth as a means of delivering cancer care and treatment in remote settings and to identify issues specific to the Territory is also required. Continued efforts to improve the facilities and treatments available, the support provided and cultural safety in care delivery will help reduce the NT’s disparities in cancer mortality and benefit Indigenous Australians and others living in remote areas.

## Figures and Tables

**Figure 1 ijerph-19-07705-f001:**
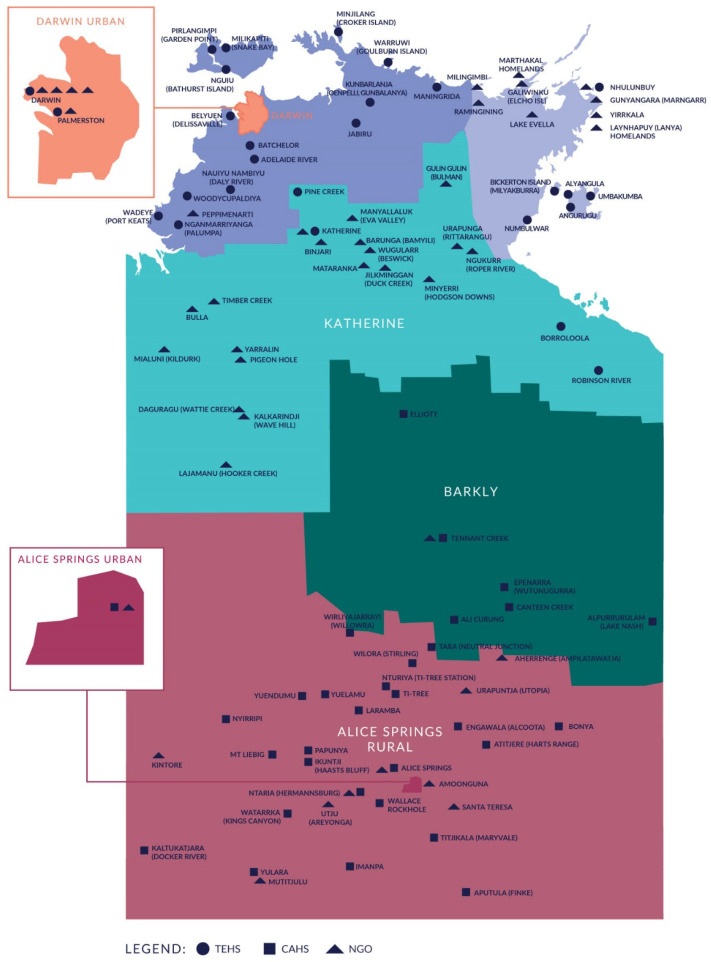
Top End and Central Australia Health Service locations [25]. TEHS = Top End Health Service; CAHS = Central Australian Health Service; NGO = Non-Government Organisation.

**Table 1 ijerph-19-07705-t001:** NT Health and Cancer Services, 2022 [25,41,42].

Regional Government Health Services	Hospitals	Estimated Population	Hospital Beds	Surgery	Radiotherapy	Chemotherapy	Palliative Care	Allied Health	Pastoral Care
Top End Health Service (TEHS)	Royal Darwin Hospital	139,000	360	✓	✓ +PET +Cyclotron	✓ (16 chairs increasing to 20)	✓ (12 bed)	✓	✓
	Palmerston Regional Hospital	37,000	116	Limited	×	×	✓	✓	✓
	Katherine Hospital	24,000 region (10,000 Katherine)	60	×	×	✓ (4 chairs)	✓	✓	✓
	Gove Hospital	16,000 region	30	×	×	×	×	✓	✓
Darwin Private Hospital			104	✓	×	✓ (4 chairs)	✓	✓	✓
Central Australian Health Service (CAHS)	Alice Springs Hospital	25,000 Alice Springs	181	✓	×	✓ (6 chairs)	✓ (10 bed)	✓	✓
	Tennant Creek Hospital	8500 region (3500 Tennant Creek)	20	×	×	×	✓	✓	✓

**Table 2 ijerph-19-07705-t002:** Availability of health workforce, per 10,000 population, 2019 [53,54].

Profession	Darwin	Central Australia	Katherine Region	Northern Territory	Australia
Aboriginal Health Practitioners	2.1	11.7	16.3	5.8	0.8
Medical Practitioners	92.8	69.7	29.3	48	205.3
Nurses and Midwives	277.3	260.9	115.8	149.6	674.5
Oral Health Practitioners	7.5	5.8	3.4	5.9	44.6
Occupational Therapists	11.9	8.4	1.9	6.5	33.3
Optometrists	1.6	1.5	np	1.3	10.6
Pharmacists	11.1	6.9	2.8	7.1	49.5
Physiotherapists	8.4	7.1	5.8	6.6	52.7
Podiatrists	1.4	2	np	1.1	9.7
Psychologists	7.2	8.4	np	6.3	53

np = not published due to low numbers.

## Data Availability

Not applicable.

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
