# Peer review of "Improving Access to Cancer Treatment Services in Australia’s Northern Territory—History and Progress"

_ijerph, 2022, doi:10.3390/ijerph19137705_

Round 1

Reviewer 1 Report

Thank you for the, appropriate, revisions.

Author Response

We thank the reviewer for their comments. 

Reviewer 2 Report

Thanks for making improvement. 

Author Response

(The authors gave the same response as above.)

Reviewer 3 Report

This is a long review paper that describes cancer care service access in Australia’s Northern Territory, a highly remote geography with a high proportion of the population being Indigenous (particularly outside of Darwin and Alice Springs).

My biggest concern is its presentation under the standard IMRaD-C format. I note in the instructions to authors that the template file does not need to be followed for review manuscripts. I would strongly believe that something like Background – Findings – Conclusion would better fit their paper and make it flow more clearly.

There currently is a Methods section, but this review doesn’t really follow a method, thus it serves no purpose. The Results section is presented under a number of subheadings, but any discussion of the implications and/or context of these is saved until the Discussion section, the latter which doesn’t have headings and jumps around between issues. These could be much more efficiently and effectively presented together, a common approach for review papers.

Overall, their review seems to be very thorough and presented appropriately against the key contexts of ‘remoteness’ and ‘Indigenous’. As per the title, there was a clear sense of the historical development (from then to now); however, I didn’t get a clear sense of how this progress rated against the ‘target’ for access. E.g. how far off is it? How much further progress is needed?

The paper’s current presentation / structure doesn’t present the findings clearly in line with the stated conclusions. I was surprised to see language like “must focus on” or “research is urgently needed into” of issues that were ‘noted’ within the paper, but I didn’t think they were clearly presented as ‘urgent’ areas to be addressed.

One ‘result’ that confused me was under 3.3, where it states that the NT had the “highest number of registered health professionals relative to its population”. This is suggestive of ‘good’ supply, but the language in the workforce section is mostly negative and (as per the conclusion) they suggest there are some urgent workforce issues to address. Please clarify how the quantitative results should be interpreted.

Author Response

Point 1: My biggest concern is its presentation under the standard IMRaD-C format. I note in the instructions to authors that the template file does not need to be followed for review manuscripts. 
I would strongly believe that something like Background – Findings – Conclusion would better fit their paper and make it flow more clearly.

There currently is a Methods section, but this review doesn’t really follow a method, thus it serves no purpose. 

The Results section is presented under a number of subheadings, but any discussion of the implications and/or context of these is saved until the Discussion section, the latter which doesn’t have headings and jumps around between issues. These could be much more efficiently and effectively presented together, a common approach for review papers.    

Response 1: We thank the reviewer for their helpful suggestions, the reviewer clearly understands what we are trying to achieve with this article. However, at the request of the Academic Editor and a previous reviewer, we have already revised this article to bring it more in line with the IMRaD-C format, including significantly expanding the Methods section. Consequently, given that three other reviewers and the Academic Editor are satisfied with the article in its current format, and the fact that we have already gone through two revisions of this article, we don’t propose at this stage to remove the Methods section (which does in fact describe the processes by which information was gathered) or make major changes to the underlying structure. 

There is considerable description included as part of the findings, and the discussion itself is fairly brief, contextualising the findings in the existing literature. 

Point 2: Overall, their review seems to be very thorough and presented appropriately against the key contexts of ‘remoteness’ and ‘Indigenous’. As per the title, there was a clear sense of the historical development (from then to now);    

Response 2: We thank the reviewer for these positive comments. 

Point 3: 
however, I didn’t get a clear sense of how this progress rated against the ‘target’ for access. E.g. how far off is it? How much further progress is needed?    

Response 3: There are seven priority areas outlined in the National Aboriginal and Torres Strait Islander Cancer Framework that align with the Optimal Care Pathway for Aboriginal and Torres Strait Islander people with cancer. These Indicators are designed to lead to better care and improved outcomes for Aboriginal and Torres Strait Islander peoples with cancer.   The Aboriginal and Torres Strait Islander Cancer Control Indicators website (https://ncci.canceraustralia.gov.au/aboriginal-and-torres-strait-islander-cancer-control-indicators) is a unique, dynamic national resource that brings together, for the first time, trusted national data to inform where our efforts can be best placed to improve Aboriginal and Torres Strait Islander cancer outcomes.
The website provides Aboriginal and Torres Strait Islander cancer data in one location designed for use by policymakers, governments, cancer organisations, researchers, health professionals, and consumers. The data on the website will enhance understanding, stimulate enquiry and inform future directions in cancer control, whether in research, policy or clinical care. It will also be a trustworthy, authoritative source of information for the broader community and people affected by cancer.

We have added in the following in the Discussion from line 508:
There are ongoing efforts to improve the reporting of Aboriginal and Torres Strait Islander cancer data as part of the implementation of the National Aboriginal and Torres Strait Islander Cancer Framework [113]. The Aboriginal and Torres Strait Islander Cancer Control Indicators website is under development [114]. It is hoped that this will enable, over time, better mapping of the extent to which efforts to improve cancer services in the NT are successfully reducing the gap in cancer outcomes for Indigenous people living the NT compared to other Territorians and Indigenous Australians living elsewhere in Australia. 

Point 4: The paper’s current presentation / structure doesn’t present the findings clearly in line with the stated conclusions. I was surprised to see language like “must focus on” or “research is urgently needed into” of issues that were ‘noted’ within the paper, but I didn’t think they were clearly presented as ‘urgent’ areas to be addressed.    

Response 4: We have toned down the language and exhortations used in some of the conclusions to better fit with the tone of rest of the paper and the steady progress documented towards improved cancer services. 

Point 5: One ‘result’ that confused me was under 3.3, where it states that the NT had the “highest number of registered health professionals relative to its population”. This is suggestive of ‘good’ supply, but the language in the workforce section is mostly negative and (as per the conclusion) they suggest there are some urgent workforce issues to address. Please clarify how the quantitative results should be interpreted.    

Response 5: Health workforce is a complex area critical to health service delivery. So, while the NT does have a higher total number of health professionals relative to the population, this seems to be offset by higher turnover than other jurisdictions. In addition, the workforce is predominantly younger, less experienced, on short-term contracts, unevenly distributed, with most of the workforce located in the urban centres, and missing key specialties and professions. 

We have added in the following at line 234 to try and make these points clearer: 
In 2018, the NT had the highest number of registered health professionals relative to its population (2,790 FTE per 100,000 people) of all Australian jurisdictions [47], however most of the health workforce is based in the capital city of Darwin and access to health professionals outside the urban centres is more limited. The NT has the lowest per capita rate of specialists and dental practitioners and the second lowest rate of allied health practitioners [47,48].

And the following in the Discussion from line 441: 
Furthermore, while the NT had a higher number of registered health professionals relative to its population, this is offset by much higher rates of turnover as well as greater numbers of staff on short-term contracts and higher use of agency-employed staff and locums than other Australian jurisdictions [85,86]. An NT study found that high turnover impacts cultural safety and continuity of care and leads to poorer health outcomes for Aboriginal peoples and higher average health costs [85]. In addition, the NT workforce overall is younger, less experienced, and unevenly distributed, with much of the workforce located in the urban centres. While increased funding has been shown to help with workforce issues in the short term, supply that was related to financial boosts tends to dissipate over time suggesting that additional strategies are required [52].

Reviewer 4 Report

Thank you for the opportunity to review this manuscript. I found this informative and very interesting. It is up to date and will inform planners and politicians as well as those recruiting health professionals in these remote areas. 

I have very minor comments 

Abstract: abbreviation NT not required

page 60 - you spend a significant time discussing the number of languages that are spoken within this reason and never come back to this point in the paper, what is the reason you have included this?

page 131- a space is required between the words so this

page 489 - you can use NT here and remove the full words 

Thank you 

Author Response

Point 1: Thank you for the opportunity to review this manuscript. I found this informative and very interesting. It is up to date and will inform planners and politicians as well as those recruiting health professionals in these remote areas.    

Response 1: We thank the reviewer for their positive comments. 

Point 2: Abstract: abbreviation NT not required    

Response2 : We have left the abbreviation NT in the abstract because it is the first time the abbreviation is used and international readers may not be familiar with the acronym. 

Point 3: page 60 - you spend a significant time discussing the number of languages that are spoken within this reason and never come back to this point in the paper, what is the reason you have included this?    

Response 3: We have previously been asked to provide more background on the context. Amidst the other description, there are three sentences discussing the number of languages that are spoken in the NT because it demonstrates the cultural and linguistic diversity of the of the NT population – and we frequently refer to the diversity of the NT population throughout the rest of the article. Furthermore, the linguistic diversity of the NT population is something that many readers may not be aware of yet is critical to understanding some of the challenges of providing care to this population 

At line 452 we have added an additional strategy for non-Indigenous staff: 
“…training on communicating with people from culturally and linguistically diverse backgrounds,”

Point 4: page 131- a space is required between the words so this    

Response 4: Fixed. 

Point 5: page 489 - you can use NT here and remove the full words      

Response 5: Changed to NT line 429. 

Round 2

Reviewer 3 Report

Thankyou for your explanation regarding the paper structure. Whilst I stand by my previous suggestions to help improve its flow and impact, I understand this is not viable for you, given conflicting views from previous reviewers and editor recommendations.

Other points are satisfactorily addressed.

This manuscript is a resubmission of an earlier submission. The following is a list of the peer review reports and author responses from that submission.

Round 1

Reviewer 1 Report

Interesting and informative work, it would be worthwhile to condense in tables or figures the data relating to the different percentages of health personnel between NT and Australia, highlighting the data on "indigenous" professionals. Is the high staff turnover related to economic aspects? Are there any indications as to why the reduction of "indigenous" health workers took place? I would suggest putting a few sentences in the discussion about this.

line 47 60% + 30% = 90% where does the remaining 10% live? The sentence is unclear. lines 66 and 68 the percentages seem contradictory

line 123-126 The sentence appears out of context: it is not clear what relevance this particular structure has with respect to the research data. It would be better to remove it or relocate it where more appropriate

Reviewer 2 Report

Thanks for the interesting paper. This overview of the cancer treatment services in the Northern Territory (NA) of Australia provides the developments from screening to cancer treatment and support for cure or palliation, which is not only relevant to the NA/Australia but also for other countries with substantial rural or remote populations or with similar histories of colonization and marginalization of the Indigenous population. It is also useful for readers to appreciate the great variations due to location/remoteness or ethnicity (indigenous or not in this case) in a developed country.

This paper focuses on the cancer treatment services at system level. I wonder if the authors can briefly describe the impact of the COVID-19 on the services. In many cases, ethnic people or people living in remote areas are most impacted - their treatment may be delayed due to shortage of staff and the public health measures in place such as restrictions. In addition, the vaccination rate may be low for the indigenous population which will exacerbate their conditions if infected. It may be too early to describe it in which case it is ok not to describe it. 

Line 413, "of' is missing. 

Reviewer 3 Report

This paper is purely descriptive. It is a comprehensive overview of cancer services and policy in the Northern Territory over recent years, which may be of interest to those interested in a description of what that’s happened and reported Policy responses. While it has sourced and extracted from reports, there is no evidence of research in the meaning of academic research as critical analysis, consideration of relevant theoretical frameworks,  and new knowledge that would make it suitable for peer review publication.